# Study of Compatibility and Flame Retardancy of TPU/PLA Composites

**DOI:** 10.3390/ma15062339

**Published:** 2022-03-21

**Authors:** Zusheng Hang, Zichun Lv, Liu Feng, Ben Liu

**Affiliations:** 1Jiangsu Key Laboratory of Advanced Structural Materials and Application Technology, Nanjing 211167, China; 2School of Materials Science and Engineering, Nanjing Institute of Technology, Nanjing 211167, China

**Keywords:** poly(lactic acid), thermoplastic polyurethane elastomer, melt blending, toughening modification, compatibility, flame-retardant properties

## Abstract

In order to apply the rigid biodegradable PLA material for flexible toothbrush bristle products, in this paper, Poly(lactic acid) (PLA) and thermoplastic polyurethane elastomer (TPU) blends (TPU/PLA composites), with a mass ratio of 80:20, were prepared by the melt-blending method to achieve toughening modification. Infrared spectroscopy, scanning electron microscopy, differential scanning calorimetry and low-field nuclear magnetic resonance were used to investigate the effect of the compatibilizer, Maleic anhydride grafted polypropylene (PP-*g*-MAH), on the compatibility of the blends, and the effect of melamine on the flame retardant properties of the blends was further investigated. The results demonstrated that 3% PP-*g*-MAH had the best compatibility effect on PLA and TPU; the TPU/PLA composites have a better macromolecular motility and higher crystallization capacity in the amorphous regions through the physical and chemical action by using PP-*g*-MAH as a compatibilizer. By adding melamine as a flame retardant, the scorch wire ignition temperature of TPU/PLA composites can reach 830 °C, which was elevated by 80 °C compared with pure PLA; however, the flame retardant effect of melamine in a single system was not significant. Melamine acts as a flame retardant by absorbing heat through decomposition and diluting the combustible material by producing an inert gas.

## 1. Introduction

Polylactic acid (PLA) is a biodegradable polymer with high tensile strength, which is presently the most promising green polymer material. However, PLA is slow to crystallize and has poor self-nucleation ability, forming a product with poor toughness that is easy to break, which needs to be modified by adding more flexible materials [1,2,3]. Polyurethane elastomer (TPU) is a polymer material, whose main structural feature is the relatively large number of urethane groups in the main chain [4,5]. These two phases together provide high strength, low temperature resistance, a high melting point and elasticity [6,7,8]. It has high toughness and elongation at the break, which can effectively improve the brittleness and processability of PLA. Melamine, as one of the important series of halogen-free flame retardants [9], has a significant effect on improving the flame-retardant performance of polyurethane, while the characteristics of melamine flame retardants, such as low smoke, non-corrosive, low toxicity [10,11], stable performance and low price, also make it widely used in the field of the polyurethane flame retardant [12,13].

TPU can be used to improve the toughening properties of PLA, by making the impact on the strength and tensile strength of composite materials significantly increased. In fact, compared to PLA, the tensile strength can be increased to 350% in the PLA/TPU blend [14,15,16]. In particular, compared with pure PLA, the impact strength of the toughened product reaches 58 km/m^2^ at a TPU content of 30%, which is a 20 times increase. In addition, the performance improvement of the blends also proved that the toughening agent has relatively high capacitive properties [17,18,19]. Finally, the SEM analysis results demonstrate that the TPU shows a uniform small spherical shape, while the blended system has a very obvious bicontinuous structure, while the impact section of the blended system has a relatively rough feature, which indicates that TPU plays a very obvious toughening effect. However, the compatibility between PLA and TPU is poor and researchers have often improved the compatibility with homemade compatibilizers. In order to obtain industrial products, such as toothbrushes with stable properties and high yields, it is necessary to select commercially available compatibilizers with stable properties and low prices. The commercial PP-*g*-MAH is one option, based on a maleic anhydride (MAH), which can be chemically linked to both PLA and TPU, respectively, to enhance the interaction [20,21].

In this paper, the effect of the compatibilizer PP-*g*-MAH on the compatibility between PLA and TPU phases in TPU/PLA blends and the flame-retardant melamine on the flame-retardant effect of the composite were analyzed in order to achieve the purpose of improving the performance of TPU/PLA blends for flexible toothbrush bristle products.

## 2. Materials and Methods

### 2.1. Raw Materials

PLA, Revode110—the weight average molecular weight (Mw) is 110 kDa, L-lactide content is −92 mol% and it was purchased from Zhejiang Haizheng Pharmaceutical Co., Ltd. The company is located in Taizhou city, Zhejiang Province, China. Thermoplastic polyurethane elastomer (TPU), WH1185—the average molecular weight (Mw) is 108.5 kDa and it was purchased from Yantai Wanhua Polyurethane Co. in China. Maleic anhydride-grafted polypropionic acid (PP-*g*-MAH)—melamine, purchased from Jiangsu Jinxiang Sairui Chemical Technology Co. The company is located in Huaian city, Jiangsu Province, China.

### 2.2. Sample Preparation of TPU/PLA Composites

TPU and PLA were dried at 80 °C for 24 h. PLA:TPU was 80%: 20 wt%; PP-g-MAH was 1 wt%, 3 wt%, 5 wt% and 7 wt%; and the corresponding melamine was 0, 10 wt%, 15 wt% and 20 wt%, which were evenly mixed. The mixed materials were added to the twin-screw extruder, and the temperature of each zone was set to 160, 170, 180, 185, 175 and 170 °C. Generally, the injection time was 6~15 s, the pressure was maintained for 5~10 s, and the pellet was cut and dried for 20~60 s before the injection molding. Melamine powder was added to each group of the test, and a silane-coupling agent was added at the same time to ensure the homogeneity of the melamine powder and pellets. After extrusion and air-cooling, the pellets were cut. The modified pellets are injection molded and drawn to produce disposable toothbrush products. The shape of the finished product is shown in Figure 1.

### 2.3. Characterization of Samples

#### 2.3.1. Infrared Spectroscopy

The testing instrument was the Nicolet IS10 Fourier infrared spectrometer. It was produced by Thermo Fisher Scientific, located in Waltham, MA, America. The test samples were made by the hot pressing into film method, where the specimens were placed on a hot press and the temperature was raised to 175 °C, and the film samples were made under a certain pressure for infrared spectroscopy testing.

The functional groups of PLA, TPU and TPU/PLA composites were analyzed by infrared spectroscopy to investigate compatibility with the formation of new chemical bonds.

#### 2.3.2. Scanning Electron Microscopy Analysis

The testing instrument was the Merlin Compact by Zeiss in Munich, Germany. The tested samples were prepared as follows: the impact-strength test specimens were punched off and the fracture surface was retained and shaped so that it could be placed flat on the test bench for the SEM morphology of the impacted section.

#### 2.3.3. Differential Scanning Calorimetry

The test instrument was a differential scanning calorimeter (DSC200F3), it was produced in Netzsch Scientific Instruments Trading Ltd., located in Waldkraiburg, Bavaria, Germany. The test method is: ramp up from 50 to 200 °C at a rate of 20 K/min, and maintain a constant temperature for 1 min to eliminate the thermal history. Three tests were conducted on each specimen.

#### 2.3.4. Low-Field Nuclear Magnetic Resonance

A PQ001-20-010V low-field NMR spectrometer from Suzhou NIUMAG Co is used for testing, located in Jiangsu, China. And dichloromethane (CH_2_Cl_2_) is selected as the solvent. The product is manufactured in Nanjing Runsheng Petrochem Co., located in Nanjing, Jiangsu Province, China. Three tests were conducted on each specimen.

#### 2.3.5. Flame Retardancy Properties

##### Vertical Combustion

The test instrument is the model CZF-5 vertical horizontal combustion tester produced by the Beijing Vertical Jinding Instrument Company, the company based in Beijing, China. Before the test, the sample strip was cut into a standard sample strip of 125 mm in length, 12.75 mm in width and 8 mm in thickness. The sample strip was clamped vertically on the sample clip of the combustion tester, and cotton was placed at the bottom (drip ignition) for flame-retardant performance testing.

##### Scorching Wire Testing

The test instrument is the SH5140 scorching wire tester produced by Guangzhou Xinhe Testing Equipment Co., located in Guangzhou, China. The sample strip was cut into a standard sample strip of 100 mm in length, 100 mm in width and 8 mm in thickness. The temperature of the burning wire was set to 700 °C, 750 °C, 800 °C, 830 °C, 850 °C and 900 °C, and the flame time was 30 s. The standard sample was placed on the sample holder, and the burning condition and self-extinguishing time were recorded.

## 3. Results

### 3.1. Compatibility Study

#### 3.1.1. Scanning Electron Microscope Analysis

Figure 2 shows the cross-sectional morphology of the composites at 5000 times magnification when the compatibilizer content is 1 wt% (a), 3 wt% (b), 5 wt% (d) and 7 wt% (e), respectively, and the surface morphology of the TPU/PLA composites shows an obvious island structure when the compatibilizer content is 1 wt%. The lack of adhesion between PLA and TPU resulted from the uninhibited aggregation of dispersed phase particles. Figure 2c shows the gaps and cavities at 1 wt% of compatibilizer, and more gaps and cavities are observed.

At a low compatibilizer content of 1%, as shown in Figure 2a, the TPU/PLA composites are characterized by a large TPU-dispersion phase domain size and a lack of adhesion between the PLA and TPU phases, which can be attributed to the lack of inhibition of aggregation between the TPU-dispersion phase particles. When the compatibilizer content was increased to 3%, as shown in Figure 2b, the TPU domain size in the TPU/PLA composites decreased and the surface was smoother. The compatibilizer improved the adhesion between the PLA and TPU phases, which was attributed to the fact that the anhydride ring of PP-*g*-MAH was opened and two carboxylic acid groups were formed by the action of the melt blending; one carboxylic acid group was esterified with the terminal hydroxyl group of PLA and the other carboxylic acid group was esterified with the end hydroxyl group of PLA. The other carboxylic acid group is attached to the carbamate in TPU to form a branched structure, forming a network structure in situ at the interface [22,23,24,25]. As can be observed in Figure 2d,e, the fracture surface becomes rough and ridged, and there is significant delamination in the direction of the fracture, showing a folded, ductile fracture feature [26,27,28,29,30,31].

However, the increasing compatibilizer content leads to the decrease of the physical properties of TPU/PLA composites, and the section reveals the presence of numerous folds. Therefore, the optimal compatibilizer content is 3%.

#### 3.1.2. Infrared Spectral Analysis

FTIR was used to analyze the molecular structure of TPU/PLA composites with different compatibilizer (PP-*g*-MAH) content to determine the chemical compatibility of the two and determine the reactions that occurred during the blending process. Figure 3 shows the IR spectra of the composites with different compatibilizer contents. As shown in Figure 3, in the IR spectra of the composites, 1740 cm^−1^ represents the absorption peak of the carbonyl-stretching vibration, 1100 cm^−1^ represents the vibration peak of the C-C skeleton; with the increase of the compatibilizer content, the carbonyl peak shows the trend of increasing and then decreasing, and the peak is strongest when the compatibilizer is 3 wt%.

Figure 4 shows the comparison of the infrared spectra of PP-*g*-MAH and TPU/PLA composites. As shown in Figure 4, PP-*g*-MAH did not show the characteristic peak at 1740 cm^−1^; thus, it can be judged that after the addition of the compatibilizer, the carbonyl group in the carbonic anhydride part of PP-*g*-MAH was chemically reacted, and the carbonyl group in the resulting -COOH changed the intensity of the carbonyl peak in Figure 4. It can be judged that PP-*g*-MAH is compatible with the TPU/PLA composites. As shown in Figure 5, the molecular structures of PLA, TPU and PP-*g*-MAH; and the compatibility mechanism of PP-*g*-MAH compatibilizers are shown respectively. Hence the best compatibility effect is at 3 wt%.

PP in PP-*g*-MAH and the matrix of TPU are physically compatible because they are both hydrocarbon macromolecules; the carboxyl group formed after the ring opening of MAH is linked with the oxygen in PLA to form an interaction to achieve the chemical compatibility.

#### 3.1.3. Differential Scanning Calorimetric Curve Analysis

Figure 6 shows the DSC curves of the composites with different compatibilizer contents, it can be observed from Figure 6 that as the temperature gradually increases, the chain segment motility of PLA molecular chains increases, and the random macromolecular chain segments in the amorphous region undergo crystal growth, producing a cold crystallization peak near 120 °C, which is the exothermic peak on the curve, and the heat absorption peak near 165 °C is the melting peak of PLA. The size of the melt peak changes with different compatibilizer contents. When the compatibilizer is 3 wt%, the melt peak is sharp and narrow, which is due to the strong interaction between PLA and TPU at this concentration, as well as the high compatibility between them and the improved crystallization ability.

Table 1 shows the DSC data of the melt crystallization of the composites. Typically, the dispersed TPU droplets in PLA matrix induce a nucleation effect on the melt crystallization of PLA, especially at low contents. On the contrary, the cold crystallization temperature of PLA shifted to a higher temperature in the presence of the TPU phase, but the degree of crystallinity improved in the case of heating with a low rate. Although the addition of TPU to PLA resulted in a decrement in onset degradation temperature, the samples degraded in wider ranges than the neat PLA [27]. From Table 1, it can be observed that the *T*cc of the TPU/PLA blends decreases and then increases with the gradual increase of the compatibilizer content, which demonstrates that PP-*g*-MAH can improve the motility of the PLA molecular chain segments and increase the possibility of regular arrangement of the molecular chains. The 3 wt% PP-*g*-MAH addition also increases the melt enthalpy of the composites from 30.703 J/g to 41.099 J/g, at which time the PP-*g*-MAH increased the motility of the composite macromolecules and the crystallinity of the amorphous region.

#### 3.1.4. Low-Field Nuclear Magnetic Resonance

The low-field NMR spectra with different compatibilizer contents are shown in Figure 7. As can be observed from Figure 7, the overall relaxation time of the TPU/PLA composite is shortest when the compatibilizer PP-*g*-MAH content reaches 3 wt%, and the relaxation time is related to the time for the system to reach equilibrium. When the compatibility of the components of the composite is good, the overall time for the system to reach equilibrium is short, which was consistent with the previous characterization.

### 3.2. Flame-Retardant Performance

#### 3.2.1. Differential Scanning Calorimetric Curve Analysis

Figure 8 shows the DSC curves of the flame-retardant content composites. It can be observed from Figure 8 that the melt peak of the composites gradually shifted towards lower temperatures as the flame-retardant content increased. Table 2 shows the thermal data corresponding to the DSC curves. From Table 2, it can be observed that the addition of the flame-retardant melamine does not have a significant effect on the *T*cc, and the addition of the melamine does not have a significant effect on the motility and alignment of the polymer molecular chains.

#### 3.2.2. Low-Field NMR

The low-field NMR spectra of composites with different flame-retardant contents are shown in Figure 9, where the peak area first decreases and then increases as the flame-retardant content increases, and the relaxation times obtained by the inversion are shown in Table 3. The relaxation time gradually decreases with the increase of the melamine addition; with the increase of melamine amount, the number of -NH groups becomes more and more, and the H proton is bound to become larger or the degree of freedom decreases.

#### 3.2.3. Vertical Burning Test

Table 4 shows the flame-retardant test results when the content of the compatibilizer PP-*g*-MAH is 3 wt% and 5 wt%, respectively, and the vertical combustion meter test results when the content of flame retardant melamine is 0, 10 wt%, 15 wt% and 20 wt%, respectively. Table 4 shows that the TPU/PLA composite does not reach the V0 level of flame retardancy when the appropriate amount of melamine is added.

PLA itself originally has an ultimate oxygen index of 21 wt%, and the level of flame retardancy is UL94HB. When it burns, a thin layer of carbonization is produced on its surface, and then the product quickly drips down and burns. After adding the melamine flame retardant, the melamine mainly exerts its flame-retardant effect by decomposing and absorbing heat and generating non-combustible gases to dilute combustible materials, but the flame-retardant efficiency of a single system of the nitrogen flame retardant is low, and only when the flame retardant reaches 20 wt% can the dripping phenomenon not occur.

#### 3.2.4. Analysis of Scorching Wire Test Results

Table 5 shows the test results of the hot wire tester when the content of the compatibilizer PP-*g*-MAH is 3 wt% and the content of the flame-retardant melamine is 0, 10 wt%, 15 wt% and 20 wt%, respectively. The table shows that when the content of the flame-retardant melamine reaches 15 wt%, the TPU/PLA composite material does not burn within 30 s of contact with the burning wire at 830 °C or extinguishes itself within 30 s after burning.

From Table 3, it can be observed that with the increase of melamine content, the temperature of the TPU/PLA composite material to withstand scorching wire is also increased, and it can withstand scorching wire contact at 830 °C for 30 s, which has a certain flame-retardant effect.

Figure 10 shows the non-combustion and post-combustion extinguishing phenomena in the hot wire test. The flame-retardant mechanism is divided into two aspects. On the one hand, melamine decomposes during combustion to produce inert gases, such as nitrogen oxide, carbon dioxide and water vapor, which can significantly reduce the concentration of volatile combustibles on the surface of the material to below the limit concentration required for combustion. On the other hand, melamine absorbs heat from the PLA matrix during the decomposition process, so that the temperature of the material drops significantly, which destroys the combustible material and heat source factors in the “fire triangle”, and finally achieves the flame-retardant effect of the material under the joint action of both.

## 4. Conclusions

In this paper, TPU/PLA composites were prepared by the melt-blending method using TPU to modify PLA, and the effect of PP-*g*-MAH on its compatibility and the effect of melamine on its flame-retardant properties were investigated. It was found that when PLA: TPU is 80:20 and the compatibilizer content reaches 3 wt%, is the best compatibility between TPU/PLA. At the same time, when the content of the melamine flame retardant reaches 15%, the PLA/TPU composite can reach the flame retardant requirement at 830 °C and has a certain flame-retardant effect.

## Figures and Tables

**Figure 1 materials-15-02339-f001:**
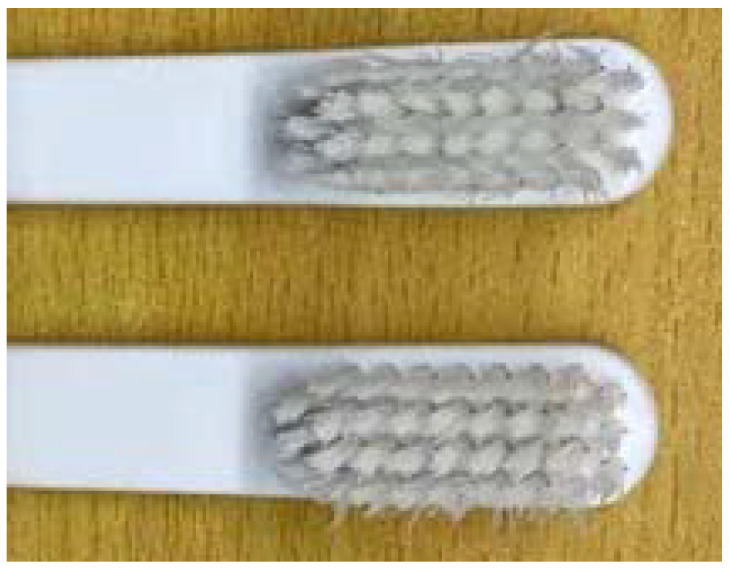
The shape of the finished product.

**Figure 2 materials-15-02339-f002:**
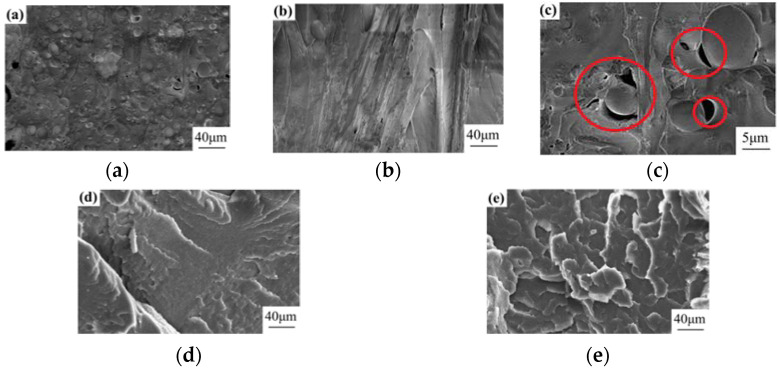
SEM photos of different compatibilizer contents. (**a**) is the morphology of the material amplified to 40 μm at 1% compatibilizer, (**b**) is the morphology of the material amplified to 40 μm at 3% compatibilizer, (**c**) is the cracks and holes in section when compatibilizer is 1%, (**d**) is the morphology of the material amplified to 40 μm at 5% compatibilizer, (**e**) is the morphology of the material amplified to 40 μm at 7% compatibilizer.

**Figure 3 materials-15-02339-f003:**
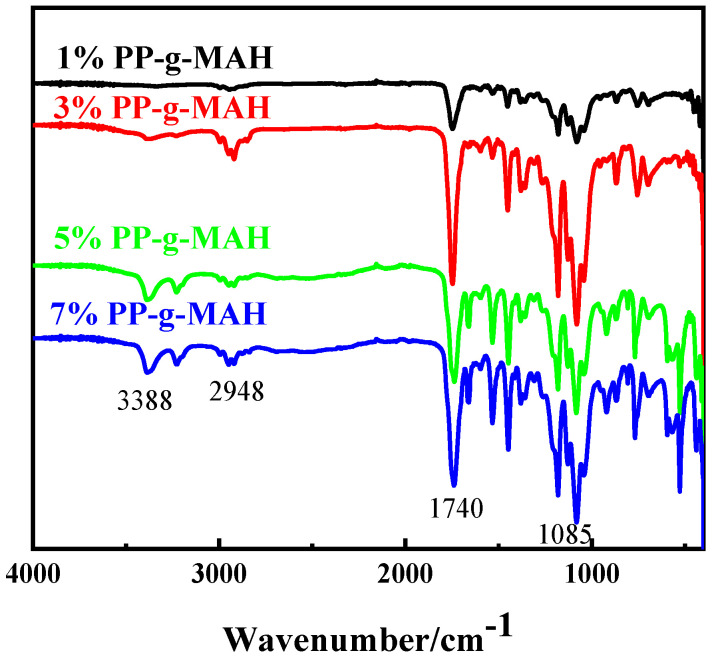
Infrared spectra of TPU/PLA composites with different compatibilizer contents.

**Figure 4 materials-15-02339-f004:**
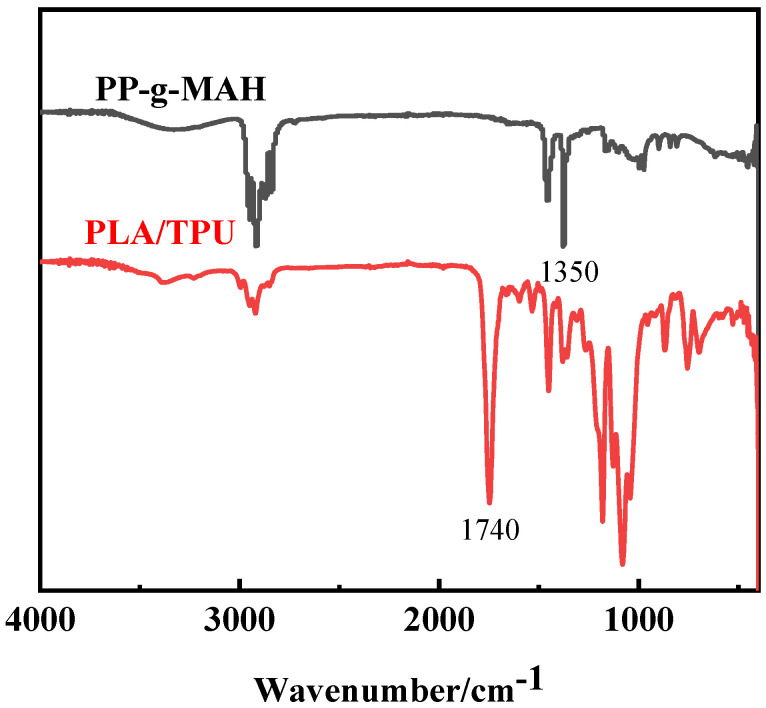
Infrared spectrum of PP-*g*-MAH and the composite.

**Figure 5 materials-15-02339-f005:**
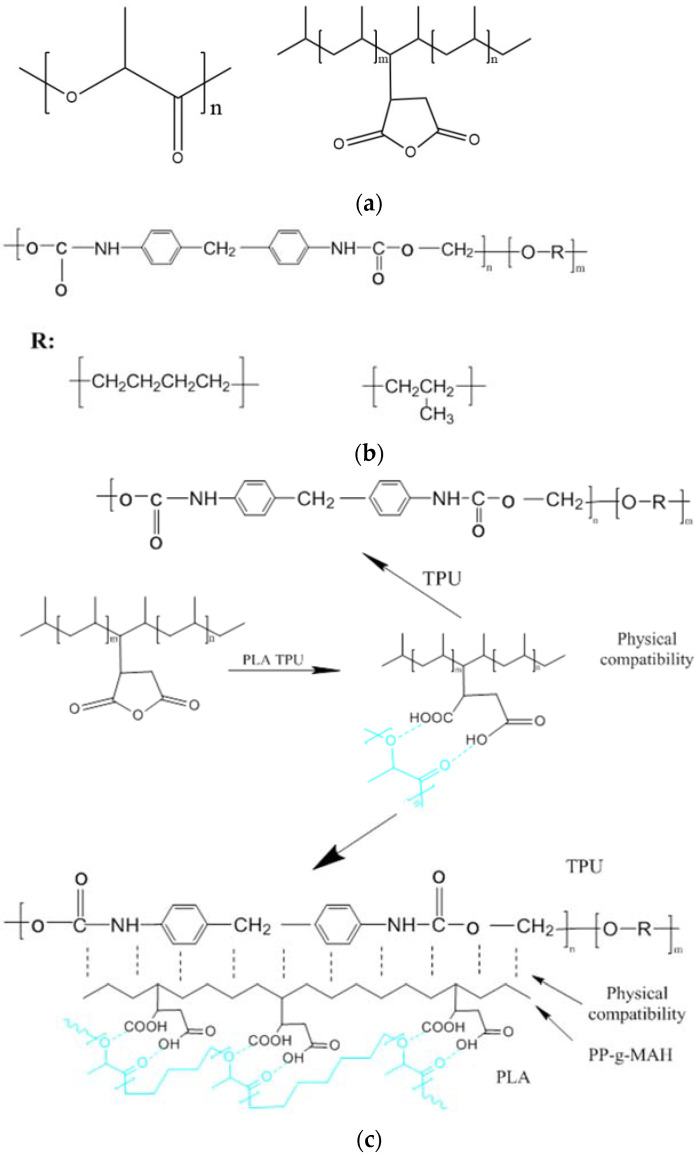
Structural representation. (**a**) Molecular structure of PLA and PP-*g*-MAH. (**b**) Structure diagram of TPU. (**c**) Compatibility mechanism of PP-*g*-MAH compatibilizer.

**Figure 6 materials-15-02339-f006:**
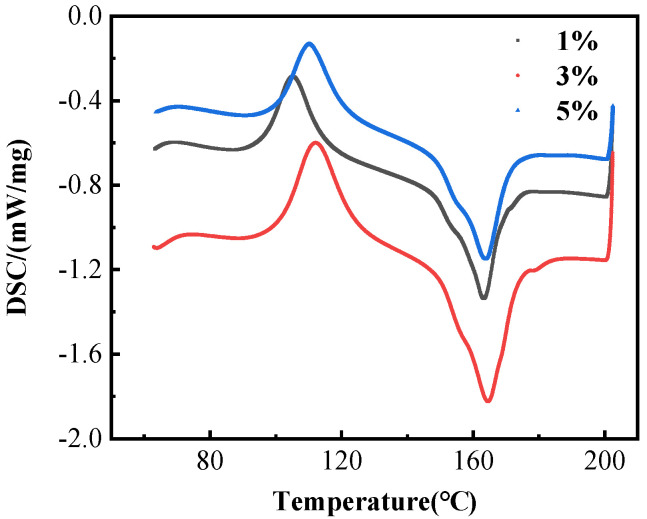
DSC curves of TPU/PLA composites with different compatibilizer contents.

**Figure 7 materials-15-02339-f007:**
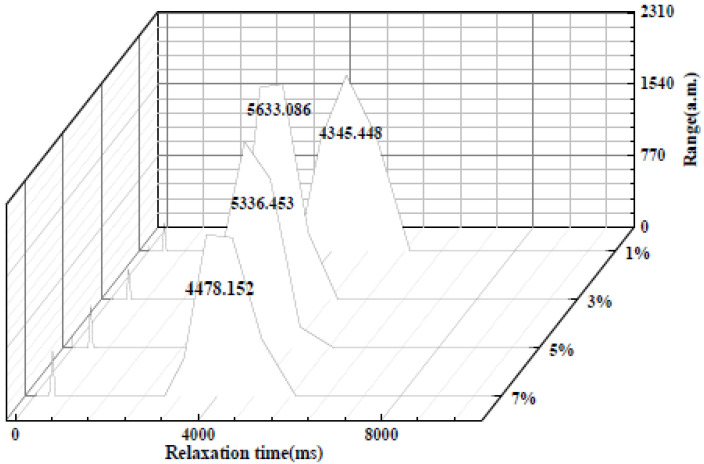
Low-field NMR spectra of TPU/PLA composites with different compatibilizer contents.

**Figure 8 materials-15-02339-f008:**
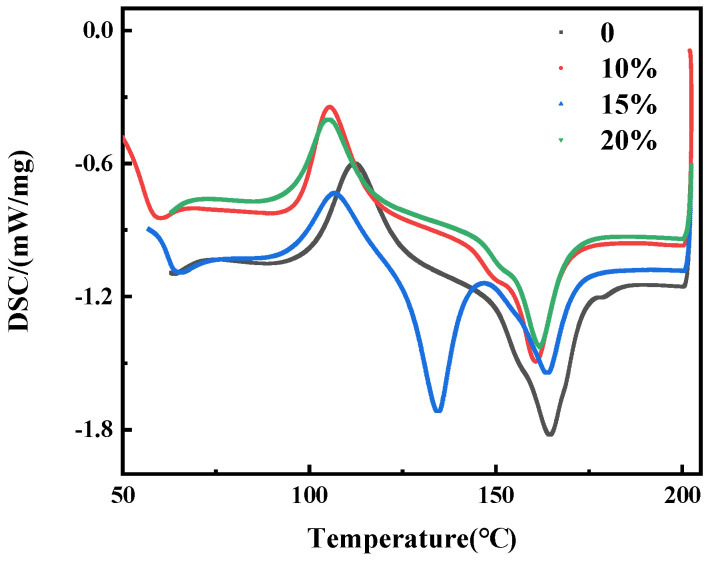
DSC curves of TPU/PLA composites with different flame-retardant contents.

**Figure 9 materials-15-02339-f009:**
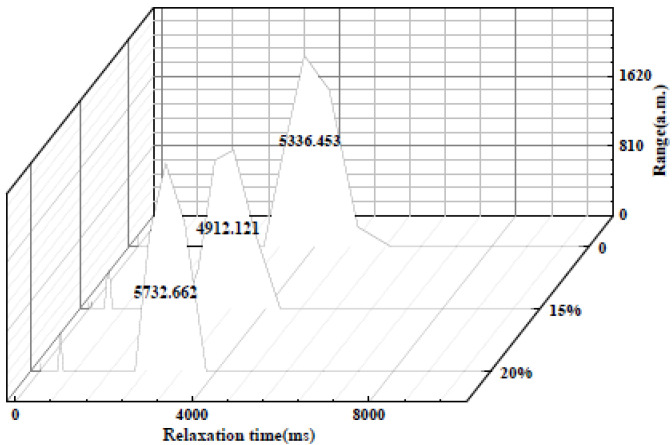
Low-field NMR spectra of TPU/PLA composites with different flame-retardant contents.

**Figure 10 materials-15-02339-f010:**
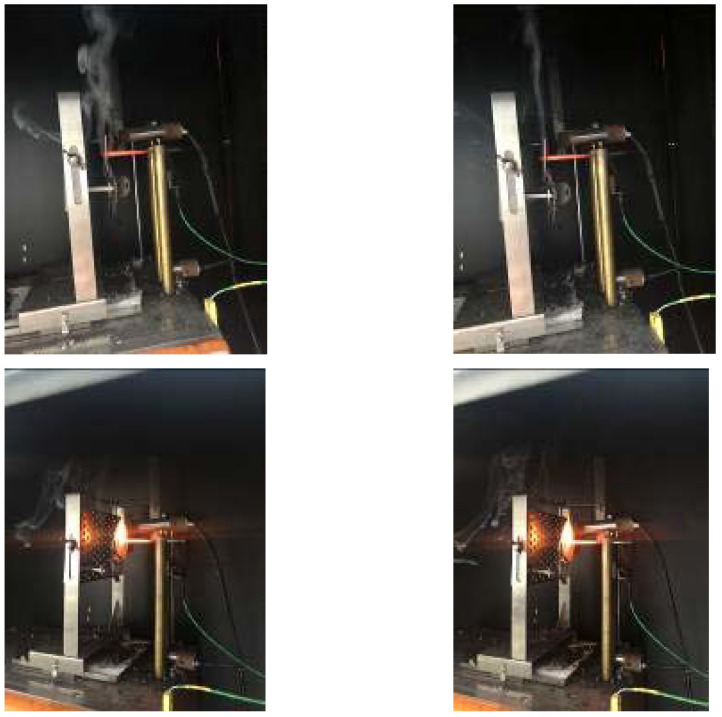
Non-combustion and extinguishing after combustion in the hot wire test.

**Table 1 materials-15-02339-t001:** DSC data of melt crystallization of TPU/PLA composites with different compatibilizer contents.

Samples	*T*cc (°C)	Δ*H*cc (J/g)	*T*m (°C)	Δ*H*m (J/g)
1 wt%	105.34 ± 0.35	13.76 ± 1.05	163.24 ± 0.54	−30.71 ± 1.11
3 wt%	102.07 ± 0.27	29.92 ± 2.23	164.57 ± 0.38	−41.09 ± 1.42
5 wt%	109.99 ± 0.33	7.76 ± 1.20	163.52 ± 0.42	−22.89 ± 0.98

**Table 2 materials-15-02339-t002:** DSC data of melt crystallization of TPU/PLA composites with different flame-retardant contents.

Samples	*T*cc (°C)	Δ*H*cc (J/g)	*T*m (°C)	Δ*H*m (J/g)
0	111.21 ± 0.55	19.14 ± 1.13	164.36 ± 0.21	−38.06 ± 2.05
10 wt%	105.28 ± 0.32	10.95 ± 1.20	160.55 ± 0.29	−21.56 ± 1.46
15 wt%	106.08 ± 0.43	19.05 ± 0.78	163.75 ± 0.48	−34.97 ± 2.13
20 wt%	105.35 ± 0.36	12.38 ± 2.21	162.22 ± 0.32	−22.57 ± 1.70

**Table 3 materials-15-02339-t003:** Peak areas of the blends corresponding to relaxation times.

Flame-Retardant Conte	Relaxation Time/ms
0	3764.94 ± 22.75
15 wt%	3274.55 ± 13.77
20 wt%	2848.04 ± 15.49

**Table 4 materials-15-02339-t004:** Vertical flammability test results.

Specimen		Combustion Phenomena	Flame Retardant Grade
Compatibility agent 3 wt%	Flame retardant 0	Ignition by dropping	V2
	Flame retardant 10 wt%	Ignition by dropping	V2
	Flame retardant 15 wt%	Ignition by dropping	V1
	Flame retardant 20 wt%	No dripping	V1
Compatibility agent 5 wt%	Flame retardant 0	Ignition by dropping	V2
	Flame retardant 10 wt%	Ignition by dropping	V2
	Flame retardant 15 wt%	Ignition by dropping	V1
	Flame retardant 20 wt%	No dripping	V1

**Table 5 materials-15-02339-t005:** Scorching wire tester test results.

Flame Retardant Conte/%	Scorching Wire Temperature/°C	Burning Phenomenon	Flame-Retardant Effect
0	750	Burning 30 s after not extinguished	Non-flame retardant
	800	Burning within 30 s not extinguished	Non-flame retardant
	830	Burning 30 s after not extinguished	Non-flame retardant
	850	Burning within 30 s not extinguished	Non-flame retardant
10	750	Not burning	Flame retardant
	800	Not burning	Flame retardant
	830	Burning 30 s after not extinguished	Non-flame retardant
	850	Burning 30 s not extinguished	Non-flame retardant
15	750	Not burning	Flame retardant
	800	Not burning	Flame retardant
	830	Extinguished after 30 s of combustion	Flame retardant
	850	Not extinguished within 30 s of burning	Non-flame retardant
20	750	Not burned	Flame retardant
	800	Not burning	Flame retardant
	830	Extinguished 30 s after burning	Flame retardant
	850	Burning within 30 s not extinguished	Non-flame retardant

## Data Availability

The data presented in this study are available on request from the corresponding author.

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
