# Peer review of "Study of Compatibility and Flame Retardancy of TPU/PLA Composites"

_materials, 2022, doi:10.3390/ma15062339_

Round 1

Reviewer 1 Report

No comments

Author Response

Thank you for your review.

Reviewer 2 Report

The study has been carefully conducted, the manuscript contains a lot of data and essential information are reported. Therefore, I recommend the publication for Materials.

Author Response

Thank you for your review.

Reviewer 3 Report

Well improved.

Author Response

Thank you for your review.

Reviewer 4 Report

The article analysis TPU/PLA compatibility and their flame retardancy. The article would be interesting but it is a bit chaotic. Therefore, I have few remarks which I hope will be taken in consideration.

1) What kind of equipment (tradename and manufacturer) was used for scanning electron microscopy test. Did the authors coat the samples with, e.g. gold or carbon?

2) Please put the scaling bars of all SEM images in the upper right corner so they could be clearly visible. Magnification of each SEM images is missing in the title.

3) I strongly suggest proofreading the article because there are some places where spaces between words are missing.

4) Figure 10. Please increase the photos because now they all look the same.

5) Regarding flame retardancy, why authors did not take into consideration cone calorimetry and LOI tests? Also, there is no discussion and any comparison with other authors works regarding flame retardancy results.

6) Conclusions missing the recommendaions, e.g. about the appropriate amount of compatibilizer and flame retardant contents.

7) It is also not clear if flame retardants were added after determination of the rational amount of compatibilizer or flame retardants were added for each compatibilizer amount. Please clearly indicate the test setup.

Author Response

This manuscript is a resubmission of an earlier submission. The following is a list of the peer review reports and author responses from that submission.

Round 1

Reviewer 1 Report

See my comment in attached file

Reviewer 2 Report

This work is potentially interesting and although the article is interesting, there are problems with the authors' style, which, in places, is a little tedious and not engaging for the reader. Therefore, the authors need to carefully read the paper and make the necessary corrections and changes to promote reader engagement. There are 11 figures in the article and I think they took half of the place in the draft. Some figures can be combined; for example, fig.8 and Fig.10 can be placed side by side. However, I feel there are more figures and tables and less explanation. I would be happy to see the TGA of  Poly(lactic acid) (PLA), TPU)  and blends(TPU/PLA composites).

Reviewer 3 Report

This is a paper on the properties of PLA/TPU with melamine as a flame retardant. Unfortunately, this reviewer cannot fin the scientific progress as well as the impact on the industrial application. The followings should be considered for their revision.

  1. The reason to pick up the material is unclear. Why do they use PP-g-MAH? As well known, PP is a brittle material and immiscible with PLA and TPU. Furthermore, using MAH is a well known method for this blend system.
  2. The detail should be written for the material. How about the molecular weight and L-lactide content for PLA? In the case of TPU, even chemical structure is not clarified. We do not know it is a polyester or polyether type. This is not good for a scientific journal.
  3. Experimental conditions should be clarified more. For example, how about the shape of injection-molded products? Injection-molded condition? Residence time in an extruder, since this is a reactive blend.
  4. In Figure 1, they said “flat” structure. I cannot understand the meaning. This must be deformed morphology by the flow field. Therefore, the authors should mention which part in the injection-molded product was employed with the indication of flow direction. Also scale bar is missing.
  5. Figure 2: Is there any possibility that the voids were created at the sample preparation (cutting by a ultramicrotome).
  6. Figure 3: Since the authors discuss the intensity, the vertical axis needs a scale. I guess the absorbance was not divided by the film thickness.
  7. Table 1-3: Mind the significant digit.
  8. English and story should be polished greatly.

Round 2

Reviewer 1 Report

No comments

Author Response

Thank you.

Reviewer 3 Report

Unfortunately, the current version was not revised enough. I believe it should be rejected.

  1. The reason to pick up the material is still unclear. PP is immiscible with PLA and TPU. Where is the PP phase? 
  2. The detail should be written for the material. Only the code name is not good enough. Need to be characterized. How about the molecular weight and L-lactide content for PLA? In the case of TPU, even chemical structure is not clarified. This is inevitable for a scientific journal.
  3. Figure 2 still can not tell the morphology. The contrast is pretty poor. Need to be stained to make a contrast. A scale bar is still unclear.
  4. FT-IR has the unit (%). 
  5. Table 1-3: Mind the significant digit. DSC is not so sensitive. I believe the authors misunderstood the question. They will understand when the error bar is introduced. Please do it.
